# Assessment of Genetic Variability for Fruit Nutritional Composition in the Ex-Situ Collection of Jujube (*Ziziphus* spp.) Genotypes of Arid Regions of India

**Vijay Singh Meena** [1,*], **Kartar Singh** [1], **Neelam Shekhawat** [1], **Rakesh Bhardwaj** [2], **Hanuman Lal** [2], **Kirti Rani** [1], **Veena Gupta** [2], **Ashok Kumar** [2], **Akath Singh** [3], **Jagan Singh Gora** [4] **and Pradeep Kumar** [3,*]

[1]   ICAR—National Bureau of Plant Genetic Resources, Regional Station, Jodhpur 342003, India
[2]   ICAR—National Bureau of Plant Genetic Resources, New Delhi 110012, India
[3]   ICAR—Central Arid Zone Research Institute, Jodhpur 342003, India
[4]   ICAR—Central Institute for Arid Horticulture, Bikaner 334006, India
[*]   Correspondence: vijay.meena1@icar.gov.in (V.S.M.); pradeep.kumar4@icar.gov.in (P.K.)

**Abstract:** Jujube or *ber* (*Ziziphus* spp.) is one of the most important fruit crops of India's arid and semi-arid regions because of its high adaptability to resource constraints hot (semi)arid region. Jujube is a rich source of minerals, vitamins and dietary antioxidants to arid zone dwellers, where it is known as poor man's apple. Given the present rising trends in discovering and exploiting plant-based health-promoting compounds, it is imperative to know the extent of variability with respect to fruit nutritional compositions present in the jujube germplasms. In this study, we assessed genetic variability for fruit nutritional and functional quality traits in eighteen jujube accessions belonging to three species (*Z. mauritiana*, *Z. rotundifolia* and *Z. oenoplia*) from an ex-situ collection from Indian arid and semi-arid regions in two consecutive fruiting seasons (2020-21 and 2021-22). Results revealed significant variability among eighteen jujube genotypes for important fruits physico-biochemical parameters. The IC 625864 (*Z. oenoplia*) identified as a superior accession for fruit antioxidant potential with having high levels of total phenols (256.2 mg/100 g dry weight) and total antioxidants (423 mg/100 g in FRAP). Moreover, IC 625849 (*Z. mauritiana*) and IC 625848 (*Z. rotundifolia*) were other genotypes containing high levels of phenols and total antioxidant (FRAP). Thus, while aiming for simultaneous improvement for total antioxidants with phenols, IC 625848, IC 625849 and IC 625864 genotypes can be considered as valuable resource for jujube fruit quality improvement program. Further, the high levels of phenotypic variance with high genotypic variance coupled with high heritability and genetic advance particularly for total antioxidants, total phenols, and ascorbic acid contents in fruits, indicating them to be considered as reliable biochemical markers to identify the productive genotypes having higher amounts of dietary antioxidants. Depending on the identified genotypes for their richness in the particular phytonutrients, these can be exploited either for table purpose or biofortification of other products, or using in jujube breeding program for quality improvement.

**Keywords:** *ber*; germplasm; *Ziziphus oenoplia*; antioxidants; arid region; phenols



## 1. Introduction

Jujube or *ber* is an important fruit crop belonging to genus *Ziziphus* of family Rhamnaceae. There are about 170 species in *Ziziphus* with huge genetic variability that are distributed across temperate to tropical regions of the globe [1]. Approximately, 20 species have been reported from India that are distributed between 8.5–32.5° N and 69–84° E geographic coordinates. Among them, *Z. mauritiana* (Indian jujube/*ber*) is the most important species which is having greatest diversity in India, distributed across tropical and subtropical regions in the country [2]. *Z. rotundifolia*, *Z. nummularia* and *Z. oenoplia* are other important species distributed particularly in (semi) arid parts of India [3].

*Z. mauritiana* L. is a highly cross pollinated and different genotypes cross freely within species, thus resulted in the build-up of a rich gene pool [3]. The maximum diversity in Indian jujube has been reported in Indian states of Rajasthan, Gujarat, Madhya Pradesh, Bihar, Punjab, Utter Pradesh, Maharashtra, Andhra Pradesh, West Bengal and Tamil Nadu [4,5]. Jujube is highly adaptive to extreme conditions of summer (45 to 47 °C) by entering into dormancy through shedding off the leaves. Besides, it is a highly drought tolerant due to deeper and strong tap root system, which extracts water from deeper soil profile in arid climates [2]. Furthermore, the leaf pubescence (as smooth or tomentose), thorniness, drooping habits as well as phytochemicals (e.g., proline, antioxidants and phenols) are important morphological and biochemical characteristics enabling its wide establishment in the resource poor semi(arid) conditions of India [2]. Moreover, it is established fact that the climate resilient crops are better performer due to their special morpho-physio-biochemical characteristics providing nutritional and environmental sustainability, and synergy [6].

The greatest challenge in present scenario is to provide nutritious food to ever increasing population, particularly those residing in hostile arid climate of India. In the western part of India, particularly of Rajasthan, per capita land availability is high but the effective days of rainfall and total annual precipitation is less, under such conditions this fruit crop can substantively play a crucial role to combat the undernourishment problem, where other major fruit crops fail to grow [2]. Therefore, jujube has recently got a commercial status in Indian arid regions as it is providing good returns to the farmers under rainfed wastelands conditions [7].

The major breeding goal, as for other commercial crops, has been to develop high yielding cultivars of jujube. But, with realization of increasingly spurring demands for high quality food, it becomes imperative to enhance the nutritional quality in a sustainable way, utilizing intra- and inter-species diversity for a specific environment using well-adapted genetic materials [6]. Jujube fruit is often referred to as the "poor man's apple" in Indian arid regions because it is relatively cheaper and rich in nutritional quality. In addition to having considerable number of health-promoting phytonutrients, jujube fruits are rich in mineral constituents such as K, P, Mn, Fe and Zn [8]. Another important jujube species collected from Indian arid regions is *Z. oenoplia* (L.) Miller and it is known as '*makai*' in Hindi and 'Jackal Jujube' in English. It is a straggling shrub and distributed particularly in the hotter parts of India, Sri Lanka, Pakistan, Malaysia and Tropical Asia [9]. Significant amounts of alkaloids, flavonoids, phenolic acids, and terpenes are reportedly present in *Z. oenoplia*, and some of these compounds have medical uses, such as the well-known cyclopeptide alkaloid (Ziziphine) [10]. Moreover, since long back it has been used traditionally in ayurvedic medicine by Indian tribes for curing of diarrhoea, fever, digestive disorders, urinary troubles, diabetes, skin infections, bronchitis, liver complaints, anaemia etc. [11].

The evaluation of the germplasm is a primary step to collect basic information of respective taxon to initiate crop improvement programs through conventional or modern breeding approaches [12]. In order to aiming at improvement of jujube nutritional and functional quality beside the economic trait, there is a need to have sufficient information about the available genetic resources rich in such traits. Moreover, the physical and biochemical traits are important markers for preliminary breeding to identify genetic variation and their role in crop improvement. Hence, evaluation of these characters becomes an imperative. Accordingly, the aim of present research was to assess variability, particularly for fruit physical traits and biochemical compounds present in the germplasm of hot (semi)arid regions of India. Information gathered or distinct germplasm identified may serve purpose of future goals of developing cultivar/variety with best possible amalgamation of higher production and phytonutrient enriched fruits.

## 2. Materials and Methods

### 2.1. Experimental Site and Plant Materials

The experiment was carried out at the National Plant Genetic Resources (NBPGR), Regional Station at Jodhpur (26°15′25.6″ N, 73°00′01.9″ E), India. In present study, the eighteen jujube accessions conserved in the field gene bank of NBPGR at Jodhpur were included, among which sixteen accessions belong to *Z. mauritiana* (Indian jujube), one genotype belongs to *Z. rotundifolia* and one belong to *Z. oenoplia* based on its diversity status (Table 1). All of the studied trees were between the ages of 30 and 36 years old and budded on *Z. rotundifolia* rootstock.

**Table 1.** Species-wise grouping of 18 accessions of jujube used in the study.

| *Z. rotundifolia* | *Z. mauritiana* | *Z. oenoplia* |
|---|---|---|
| IC 625848 | IC 625849, IC 625850, IC 625851, IC 625852, IC 625853, IC 625854, IC 625855, IC 625856, IC 625857, IC 625858, IC 625859, IC 625862, IC 625863, IC 625870, IC 625871 | IC 625864, IC 625869, |

The experimental field's soil type was sandy loam, with a pH of 7.95, an organic carbon content of 0.25 percent, an electrical conductivity of 0.24 dS m$^{-1}$, and accessible phosphorus and potassium concentrations of 47 kg ha$^{-1}$ and 573.75 kg ha$^{-1}$, respectively. As a recommended cultural practice, 30 kg well rotten farm-yard manure and 500 g diammonium phosphate were applied to tree basin and mixed at the onset of monsoon every year. Pruning was undertaken during mid-May as standard practice for quality fruit production in current season shoot growth. Weeding and hoeing were performed during July to October as per the need, and irrigation was given when prolong dry spell occurred.

### 2.2. Determination of Physico-Biochemical Parameters of Fruits

Twenty randomly sampled mature fruits in each tree (replication) for each accession were harvested and used for determinations of fruit physical and biochemical quality traits. Fruit related parameters namely edible portion (%), moisture (%), ascorbic acid, protein, sugar, total antioxidants—Ferric Reducing Antioxidant Power (FRAP), ash, total soluble solids (TSS, °Brix), total phenols, fruit length, fruit width and fruit weight, were recorded in all eighteen jujube accessions.

Fresh fruits with uniform size without signs of defect and visual injuries were harvested at edible maturity for different physico-biochemical analysis in both the years. Average fruit weight was measured by electronic weighing machine (Ohaus, Shanghai, China) and fruits length and fruits width was measured with the help of digital vernier caliper (Mitutoyo, Kanagawa, Japan). The proximate composition of 18 accessions was determined using the recommended methods (AOAC 2005) viz., moisture (AOAC 934.01), ash (AOAC 938.08), dietary fiber (AOAC 985.29) and protein (AOAC 2001.11). Sample preparation for total sugar, total phenols and total antioxidants was done as described by Arivalagan [13]. The TSS contents were recorded with the help of hand refractometer (0–32 °Brix; Erma, Tokyo, Japan). Ascorbic acid content (vitamin C) was estimated by titration method using 2,6-dichlorophenol indophenol dye solution [14]. In addition to three biological replicates, the biochemical analysis was done using three technical replicates.

Total phenols or phenolic content was estimated spectrophotometrically using Folin–Ciocalteu reagent [15]. Briefly, 100 μL of the sample extract (80% ethanol), 2.9 mL of deionized water, 0.5 mL of Folin–Ciocalteu reagent and 2.0 mL of 20% Na$_2$CO$_3$ solution were added. The mixture was allowed to stand for 90 min and absorption was measured at

760 nm against a reagent blank in UV–vis spectrophotometer (Shimadzu, Tokyo, Japan) and results were expressed as gallic acid equivalent (mg GAE/100 g).

Total antioxidants activity was determined in term of Ferric reducing antioxidants power (FRAP) according to the procedure described by Benzie and Strain [16]. Briefly, the FRAP reagent included 300 mM acetate buffer, pH 3.6, 10 mM TPTZ in 40 mMHCl and 20 mM FeCl$_3$ in the ratio 10:1:1 ($v/v/v$). Three ml of the FRAP reagent was mixed with 100 μL of the sample extract in a test tube and vortexed in the incubator at 37 °C for 30 min in a water bath. Reduction of ferric-tripyridyltriazine to the ferrous complex formed an intense blue color which was measured; at a UV–vis spectrophotometer at 593 nm at the end of 4 min. Results were expressed in terms of FRAP mg/100 g.

### 2.3. Statistical Analysis

The experiment was carried out in a randomized block design with three replications. One tree for each of eighteen accessions (treatments) was planted randomly in each of three blocks (replications). Twenty randomly sampled fruits per tree (replication) were collected for physico-biochemical analysis. Statistical analysis was done as per the procedures given by Panse and Sukhatme using Genstat statistical software [17]. Estimation of variability, heritability and genetic advance was carried as given by Burton and DeVane [18]. Variance and simple correlation coefficients were analyzed using PAST software, ver. 3.25 [19]. Correlation analysis among different phytochemicals was performed using the 'psych' package in R (R statistical software V.4.2.1 (R core team 2021) [20].

## 3. Results and Discussions

### 3.1. Variations among Germplasms for Different Fruit Morphometric Traits

The mean values of fruit physical traits presented in Figure 1 shows a wide variation among 18 jujube genotypes for fruit length, width and weight. Fruit length and width differed by 7.61 and 5.17-fold and varied from 9.02–70.05 (mm) and 11.2–57.9 (mm) with a mean value of 31.7 mm and 25.5 mm, respectively. The fruit weight (g) differed by 38.4-fold and varied from 1.4 to 53.8 with the mean value of 14.6 (Figure 1). The jujube genotype, IC 625869 (*Z. mauritiana*) had distinctly higher values for fruit-length, -width and -weight than rest of the other genotypes. In contrast, genotype IC 625864 (*Z. oenoplia*) noted with lowest fruit-weight and -size (length and width), that was close to *Z. mauritiana* (IC 625849) and *Z. rotundifolia* (IC 625848) accessions. The variations in fruit physical traits in the accessions within the jujube species have been adequately reported in the studies from arid as well as other regions [21,22], but variations at species levels are scarcely available particularly from Indian hot-arid regions. Considering the fact that the distinctive phenotypic traits of fruits in the germplasms are controlled primarily by four major QTLs (genetic maternal) like *globe*, *sun*, *ovate*, and *fs8.1*. The *globe*, had a greater unique effect on fruit length than fruit width, and this is likely the primary factor driving changes in fruit size [23,24]. Thus, jujube fruit length may be shorter than width, depending on fruit shape, which can range from obovate/round/oval to oblong.

### 3.2. Estimation of Genetic Variability for Different Biochemical Parameters

The mean square values of important fruit biochemical parameters are presented in Table 2. Significant genotypic differences ($p \leq 0.001$) were recorded among the genotypes for all the biochemical compounds studied. This showed that all the quality parameters differed widely and very significantly among 18 jujube accessions. Effect of environment was evident for some traits viz., edible portion, moisture, protein, sugar and ash contents, while it had no significant effect on the fruits' bioactive compounds like phenols, antioxidants, ascorbic acid and sugar (Table 2).

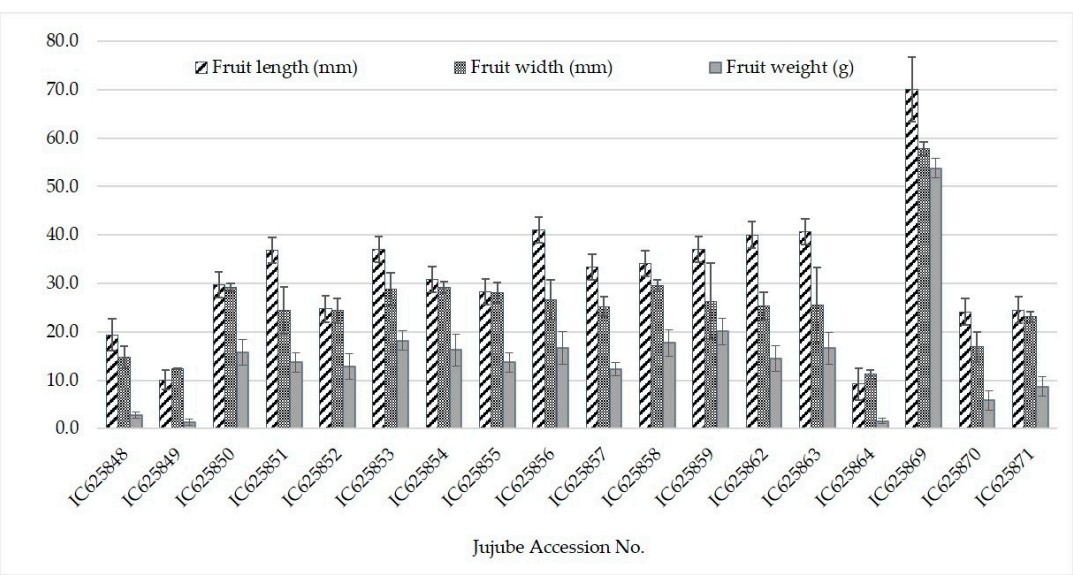

**Figure 1.** Physical fruit characteristics (fruit length, mm; fruit width, mm; and fruit weight, g) of jujube fruits (*n* = 40) for each of the 18 genotypes of jujube tree studied in the two-year study. Mean values ± sd, *n* = 40.

**Table 2.** Mean sum of squares for different biochemical parameters of the fruits of the 18 genotypes of jujube studied during two-years of study.

| Source of Variations | df | Mean Sum of Squares | | | | | | | |
|---|---|---|---|---|---|---|---|---|---|
| | | Edible Portion (%) | Moisture (%) | Ascorbic Acid (mg/100 g) | Protein (%) | Phenols (mg/100 g) | Sugars (%) | Antioxidants (mg/100 g) | Ash (%) |
| Replication | 2 | 86.03 | 8.74 | 39 * | 0.111 | 181 | 0.40 | 2156 | 0.0165 |
| Genotype (G) | 17 | 245.43 *** | 170.29 *** | 4934 *** | 1.500 *** | 28,305 *** | 11.86 *** | 125,341 *** | 0.2002 *** |
| Environment (E) | 1 | 123.10 ** | 130.36 *** | 0 | 11.561 *** | 0 | 74.25 | 4 | 1.1824 *** |
| G × E | 17 | 2.17 | 2.19 | 0 | 0.398 | 0 | 0.66 | 0 | 0.1464 ** |
| Error | 70 | 11.50 | 10.76 | 12 | 0.416 | 11 | 1.36 | 13 | 0.0604 |

*, **, *** Significant at *p* < 0.05; < 0.01; < 0.001, respectively.

The mean values reported in Figure 2 showed a wide variation in percent fruit ash and fruit moisture contents in 18 jujube accessions, indicating diversity in inorganic and organic compounds present in their fruits. The highest ash content (%) differed by 7.09-fold and varied from 0.22 to 1.56% with the mean value of 0.61%. The genotypes IC 625849, IC 625848 and IC 625871 contained higher fruit ash, while IC 625870, IC 625850, IC 625851 and IC 625854 genotypes had lower contents. The values for the fruit moisture content (%) and ash content (%) presented in bar diagrams in Figure 3 showed inverse trend with each other. Previous reports state that the moisture content in food is concerned with the amount of water present; and ash content in food can be an inorganic residue remaining after either ignition or complete oxidation of organic matter, and rich in specific inorganic minerals [25,26]. Therefore, the fruit with higher ash contents in our study might also have higher level of mineral contents besides having some important organic compounds. However, since different foods have different compositions, so a direct relationship may not be expected but, within one type of food, the higher the dry matter, the higher the ash such as in jujube. Hence, it can be inferred that the higher the fruit moisture content, the lower the fruit dry matter content (ash), or vice -versa, this relationship has been reported by several researchers [2–30]. Furthermore, the presence of free moisture is directly related to water activity; and an important indicator of shelf life [31–33]; while the ash content is reportedly linked to a food's nutrition and longevity [34–36]. Hence, the moisture and ash content are important determinants of aesthetics and nutritional values of fruits.

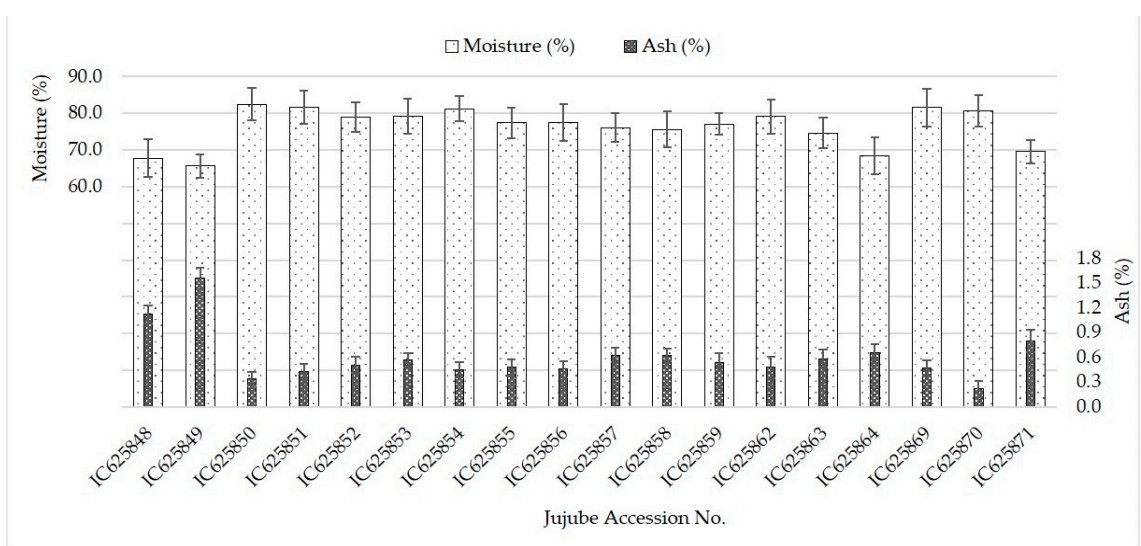

**Figure 2.** Mean values for the moisture content (%, *Y* axis, left) and ash content (%, *Y* axis, right) of jujube fruit (*n* = 40) for each of the 18 genotypes of jujube tree studied in the two-year study. Mean values ± sd, *n* = 40.

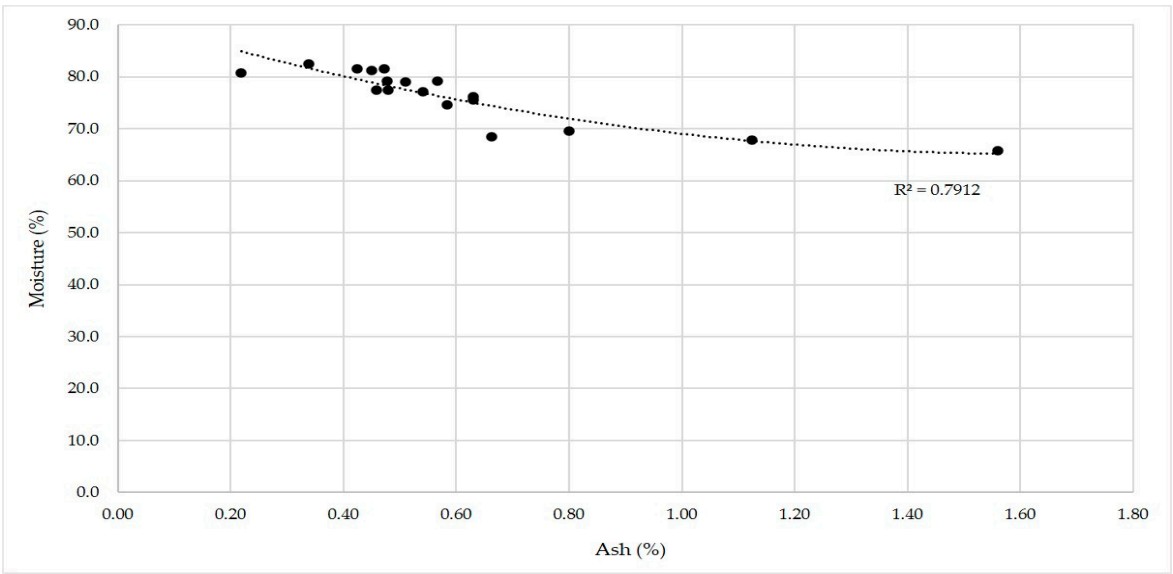

**Figure 3.** Relation between fruit moisture content (%) and ash content (%) of the fruits of the 18 genotypes of jujube tree studied. (Mean values, *n* = 40).

The mean performance, range, coefficient of variation (CV) and least significant differences (LSD) presented in Table 3 also showed a sufficient amount of variation for phytochemical compounds in studied jujube genotypes. The pooled data of two-year analysis for different biochemical parameters revealed a high range of variability among 18 jujube accessions with a high statistical difference ($p \leq 0.05$). The measured quality traits like TSS ranged from 12.5 to 24.2 °B, ascorbic acid from 28.7 to 139.1 mg/100 g, total phenols from 13.1 to 256.1 GAE/100 g, and total antioxidants from 22.0 to 423.2 mg/100 g (Table 3). The TSS content of fresh fruit (°B) differed by 1.93-fold among jujube accessions, and ranged from 12.5 to 24.2 (average, 19.8). Its content was high in IC 625863, IC 625852, IC 625850, IC 625870 and IC 625871 and low in IC 625859, IC 625848, IC 625864 and IC 625858. The ascorbic acid content (mg/100 g) ranged from 24.0 to 138.7 among genotypes (average, 59.6). The maximum concentration was assayed in IC 625849 followed by IC 625871, IC 625863;

whereas it was lowest in IC 625851, IC 625858 and IC 625853. For several characteristics of fruit quality, shelf life, processing, carbohydrate metabolism, and biotic and abiotic stressors, the total soluble solid content in the fruit is crucial. Furthermore, sugar is the major constituent of TSS and an important determinant of fruit quality for processing [37]. The TSS content of fruit shows a linear relationship with dry matter content, which was maintained from harvest to senescence, and helps extend fruit storage period, thus can be used as indicators of fruit shelf life [38]. The ascorbic acid is considered as an important antioxidant and plant growth regulator, and acts as an anti-stress agent [39]. The significant roles of ascorbic acid are proven in many fruit trees in scavenging/chelating the free radicals and activating the natural defense against many stresses [40–42]. Ascorbic acid also acts as a co-factor in biosynthesis of ethylene, gibberellins and abscisic acid phytohormones, and playing an important regulatory role in many physiological (cell division and differentiation) and biochemical processes (antioxidant and quality traits) of fruit plants [42]. The jujube accession IC 625863 and IC 625849 can be considered ideal for processing industry since they contained higher TSS (24.2°B) and ascorbic acid (139.2 mg/100 g), respectively. These genotypes can be further used in jujube breeding in selection/hybridization program to develop cultivar with desired traits. Other reports also reveal considerable variations in TSS content among different jujube genotypes [21,43].

**Table 3.** Variations in mean values of physico-biochemical parameters (fresh weight basis) determined for the fruits of 18 genotypes of jujube for the period of 2020-21 and 2021-22 (*n* = 20).

| Treatments | Total Soluble Solids (°B) | | | Ascorbic Acid (mg 100 g$^{-1}$) | | | Total Phenols (mg GAE 100 g$^{-1}$) | | | Total Antioxidants (FRAP, mg 100 g$^{-1}$) | | |
|---|---|---|---|---|---|---|---|---|---|---|---|---|
| | 2020-21 | 2021-22 | Pooled | 2020-21 | 2021-22 | Pooled | 2020-21 | 2021-22 | Pooled | 2020-21 | 2021-22 | Pooled |
| IC 625848 | 12.83 | 13.37 | 13.10 | 90.00 | 90.16 | 90.08 | 155.30 | 155.63 | 155.47 | 364.55 | 365.22 | 364.88 |
| IC 625849 | 16.47 | 16.40 | 16.43 | 139.24 | 139.00 | 139.12 | 163.41 | 163.00 | 163.20 | 486.71 | 487.30 | 487.00 |
| IC 625850 | 23.33 | 22.77 | 23.05 | 67.93 | 69.00 | 68.47 | 18.41 | 18.19 | 18.30 | 40.71 | 42.37 | 41.54 |
| IC 625851 | 20.00 | 20.97 | 20.48 | 28.34 | 29.00 | 28.67 | 12.86 | 13.34 | 13.10 | 21.38 | 22.62 | 22.00 |
| IC 625852 | 23.70 | 23.90 | 23.80 | 64.17 | 61.67 | 62.92 | 29.58 | 30.21 | 29.90 | 49.77 | 67.33 | 58.55 |
| IC 625853 | 21.33 | 21.83 | 21.58 | 54.30 | 53.00 | 53.65 | 30.06 | 30.06 | 30.06 | 98.43 | 95.97 | 97.20 |
| IC 625854 | 21.40 | 22.00 | 21.70 | 61.06 | 62.00 | 61.53 | 26.09 | 26.13 | 26.11 | 79.87 | 81.47 | 80.67 |
| IC 625855 | 20.00 | 20.57 | 20.28 | 61.71 | 62.67 | 62.19 | 28.48 | 28.87 | 28.67 | 80.31 | 81.98 | 81.14 |
| IC 625856 | 20.47 | 22.47 | 22.48 | 89.65 | 89.66 | 89.65 | 26.69 | 26.99 | 26.84 | 77.86 | 80.19 | 79.03 |
| IC 625857 | 20.67 | 18.17 | 19.42 | 76.49 | 77.08 | 76.79 | 14.96 | 15.31 | 15.14 | 32.63 | 34.29 | 33.46 |
| IC 625858 | 15.00 | 14.00 | 14.50 | 52.24 | 52.79 | 52.52 | 15.15 | 15.33 | 15.24 | 30.22 | 30.89 | 30.56 |
| IC 625859 | 13.00 | 12.00 | 12.50 | 71.03 | 71.33 | 71.18 | 17.87 | 18.10 | 17.98 | 28.04 | 30.69 | 29.36 |
| IC 625862 | 21.87 | 21.00 | 21.43 | 82.68 | 83.68 | 83.18 | 30.69 | 30.95 | 30.82 | 93.03 | 92.68 | 92.85 |
| IC 625863 | 24.40 | 24.00 | 24.20 | 121.58 | 122.67 | 122.12 | 45.86 | 46.13 | 46.00 | 126.21 | 135.21 | 130.71 |
| IC 625864 | 14.00 | 14.20 | 14.00 | 66.43 | 50.76 | 58.59 | 256.36 | 256.05 | 256.20 | 422.41 | 424.07 | 423.24 |
| IC 625870 | 23.67 | 23.23 | 23.45 | 106.35 | 106.35 | 106.35 | 20.11 | 20.56 | 20.33 | 51.23 | 52.90 | 52.07 |
| IC 625871 | 22.97 | 23.23 | 23.45 | 125.27 | 125.67 | 125.47 | 42.88 | 42.55 | 42.72 | 128.06 | 129.73 | 128.90 |
| Mean | 19.71 | 19.65 | 19.76 | 79.91 | 79.21 | 79.56 | 54.99 | 55.14 | 55.06 | 130.08 | 132.64 | 131.36 |
| Range | 12.8–24.4 | 12.0–24.0 | 12.5–24.2 | 52.2–139.2 | 29.0–139.0 | 28.7–139.1 | 12.9–256.4 | 13.3–256.1 | 13.1–256.1 | 21.4–422.4 | 22.6–424.1 | 22.0–423.2 |
| CV | 4.15 | 3.93 | 1.24 | 3.3 | 8.09 | 3.16 | 1.75 | 1.77 | 0.43 | 6.58 | 4.8 | 2.19 |
| LSD at 5% | 1.40 | 1.20 | 0.50 | 4.00 | 10.80 | 5.30 | 1.70 | 1.70 | 0.50 | 14.10 | 10.50 | 6.00 |

　　　　Further, data presented in Table 3 showed that, the contents of total phenols (mg 100 g$^{-1}$ FW) differed by 19.2-fold among the accessions from 13.64 to 256.2 (average, 55.06). Its content was highest in IC 625864 followed by IC 625848 and IC 625849, while lower content was recorded in IC 625851, IC 625857 IC 625858 and IC 625859. The jujube accession IC 625864 belonging to *Z. oenoplia* was found superior for total phenols content (256.2 GAE g$^{-1}$ FW). Unlike other jujube accession, it bears smaller fruits (average 1.6 g) which is round and turned black during ripening. The variation in the total phenols in the fruits of jujube cultivars of China ranged from 5.18 to 8.53 mg GAE g$^{-1}$ [44]. Variation in contents and range in the total phenols between the two studies may be due to different genotypic background of jujube accessions, mostly arising from natural cross pollination and self-incompatibility, besides different growing conditions, and methods of analysis among other factors.

The total antioxidant contents (mg 100 g$^{-1}$ FW) ranged widely, from 22.0–487.0 (average 131.4), showing 22.13-fold difference. Further, the presence of substantial amounts of phenols in Indian jujube indicates that they are a significant source of antioxidants which may provide health promoting benefits. The content (μmol g$^{-1}$) of total antioxidants was recorded highest in accessions IC 625849 (487.0), IC 625864 (422.0) and IC 625848 (364.2). The phenolics compounds are ubiquitous secondary metabolites and act as a natural source of antioxidants in plants [45]. It is extensively accepted that the antioxidants capacity of extracts obtained from plants is attributed to total phenolic content [46]. The genetic makeup of the fruits essentially determines their quality, and environmental factors over the cropping seasons or years have little or no impact on high heritable traits. The insignificant variations for TSS, ascorbic acid, total phenols and antioxidants contents between the two years show that these traits were less influenced by environments (Table 3). Cervantes et al. [47] have also reported similar findings in strawberry while evaluating different cultivars. Natural antioxidants produced from natural sources have received a lot of interest recently since they are safer and have less adverse effects than synthetic antioxidants [48–51]. Similarly, the antioxidants help alleviating cellular damage by limiting reactive oxygen species (ROS) [52–56]. These compounds have been well established for scavenging the lethal effect of ROS and thereby protecting our body against cardiovascular disorder, cancer and other diseases [53,55]. The significant variation for various quality traits and antioxidants in jujube and other crop species were reported in earlier studies [22,56,57].

Scientific studies reveal that *Z. oenoplia* species is rich in alkaloid of cyclopeptide called as Ziziphine [10] and has been used in Indian medicine system for the treatment of diseases like ulcer, stomach ache, obesity, asthma, diarrhoea, fever, bronchitis, liver complaints, anaemia etc. The present study also confirms richness of IC 625864 accession of *Z. oenoplia* in total phenols and total antioxidants contents. The findings of this study revealed a wide variation among biochemical attributes that show huge potential of the studied germplasm. It can be used in future breeding programs to improving productivity and quality traits fruits. Moreover, the variability observed in the current study could also pave a way to future pharmacology studies. Overall, jujube can be considered a good source of natural antioxidants, and can be used potential future fruit to combat malnutrition problems in developing countries like India.

For understanding the extent of genetic or environment factors governing the expression of fruit quality traits in the diverse jujube accessions, we studied different components of genetic analysis viz., genotypic-, phenotypic-, and environmental-variance (Vp, Vg and Ve), and phenotypic and genotypic coefficient of variation (PCV and GCV), heritability and genetic advance, as well as correlation matrix in two successive fruiting seasons. The genotypic variances were recorded highest for total antioxidants followed by total phenols and ascorbic acid; while these were lowest in the ash, protein, sugar and TSS (Table 4). The degree of PCV was slightly higher than the corresponding GCV for all the biochemical traits, except TSS content, indicating effect of environment on the expression of phenotype, whereas the prominently low difference between PCV and GCV was recorded for total antioxidants, total phenols and ascorbic acid, indicating less influence of environmental factors on the expression of these traits. This fact was also supported by the recorded high heritability (>90%) for the traits related to fruit antioxidants potentials (Table 5), thereby implying these traits to be genetically more stable. The genetic advance (GA) as a percent of mean, ranged from 8.99 to 229.6 percent, was on higher side for total antioxidants, total phenols and protein, thus confirming them more stable quality traits, hence liable to be used as biomarker in jujube quality improvement program.

**Table 4.** Values of variance and coefficients of variation of the fruit quality parameters determined for the fruits of 18 Jujube genotypes in the two-year study period.

| Fruit Quality Parameters | Genotypic Variance | Phenotypic Variance | Environmental Variance | GCV | PCV |
|---|---|---|---|---|---|
| Total Soluble Solids (°B) | 3.76 | 17.57 | 13.82 | 9.44 | 20.41 |
| Ascorbic acid (mg 100 g$^{-1}$) | 823.55 | 834.45 | 10.90 | 34.79 | 35.02 |
| Total Phenols (mg 100 g$^{-1}$) | 4266.10 | 4276.38 | 10.28 | 111.33 | 111.46 |
| Total Antioxidants (mg 100 g$^{-1}$) | 17,513.30 | 17,523.82 | 10.53 | 102.42 | 102.45 |
| Sugars (%) | 3.43 | 3.82 | 0.39 | 58.80 | 62.03 |
| Protein (%) | 1.60 | 1.67 | 0.08 | 87.91 | 89.99 |
| Moisture (%) | 42.95 | 59.41 | 16.46 | 8.81 | 10.36 |
| Ash (%) | 0.13 | 0.15 | 0.02 | 50.92 | 53.74 |

GCV—genotypic coefficient of variation; PCV—phenotypic coefficient of variation.

**Table 5.** Estimates of heritability, genetic advance (GA) and GA as percentage of the mean for fruit quality parameters determined for the fruits of 18 jujube genotypes in the two-year study period.

| Fruit Quality Parameters | Heritability (%) | GA | GA as % of Mean |
|---|---|---|---|
| Total Soluble Solids (°B) | 21.37 | 1.85 | 8.99 |
| Ascorbic acid (mg 100 g$^{-1}$) | 98.69 | 58.73 | 71.21 |
| Total Phenols (mg 100 g$^{-1}$) | 99.76 | 134.39 | 229.06 |
| Total Antioxidants (mg 100 g$^{-1}$) | 99.94 | 272.53 | 210.91 |
| Sugars (%) | 89.87 | 3.62 | 114.83 |
| Protein (%) | 95.43 | 2.54 | 176.90 |
| Moisture (%) | 72.29 | 11.48 | 15.44 |
| Ash (%) | 89.79 | 0.71 | 99.40 |

The total antioxidants, total phenols and ascorbic acid contents had a greater amount of genetic variance and genotypic coefficients of variation, and therefore, there is a good possibility and potential for improvement in jujube genotypes through hybridization and selection methods of breeding. Furthermore, the lower values of GCV and PCV were reported for moisture %, TSS, ash and sugar contents. The similar results were reported for ascorbic acid in cabbage head and tomato [58–61]. Therefore, a reliable selection can be made for these traits. The effectiveness and potentiality of the traits under selection could be revealed by an assessment of genetic gain. The estimation of heritability and genetic advance are very important selection parameters to find out the heritable portion of variability and genetic gain which is to be achieved in the next generations. The high heritability along with high genetic advance was recorded for total antioxidants (mg/100 g) and total phenols (mg/100 g) among the jujube accessions. The parallelism of the magnitude of heritability and degree of genetic gain happens due to predominant role of additive gene [62]. These findings are consistent with previous studies [59,61]. It is believed to be controlled by additive genes and phenotypic selection for their improvement could be accomplished through simple breeding methods. Meanwhile, it is not necessarily true that high heritability would always exhibit high genetic advance. High heritability along with low genetic advance was recorded for protein, sugar and ash %. These combinations of genetic parameters might be attributed to non-additive gene action for the control of these characters [62]. Earlier researches have also suggested heterosis breeding for the improvement of total carotenoid in petals of marigold (*Tagetes erecta*), and ascorbic acid and antioxidants enzymes in cabbage head [63–65]. Therefore, improvement could be made through heterosis breeding. Similar findings were also reported and hence, early generation selection would be helpful for improving these characters [66,67].

### 3.3. Genetic Correlations

Early studies on correlation coefficients and the magnitude and direction of the genetic correlations (positive and negative) among traits help in selective decisions and subsequent planning of breeding strategies. A strong positive correlation between ascorbic acid (ascorbic acid) and total antioxidants (r = 0.79); total phenols and total antioxidants (r = 0.94) are evident in the present study (Figure 4). The IC 625849 identified as a potent accession rich in ascorbic acid with total antioxidants, while accessions, IC 625864, IC 625848, IC 625849 having high level of phenols with total antioxidants. A significant positive association of phenols with ascorbic acid (r = 0.72) and protein (r = 0.69) were also reported. This suggests a possibility of simultaneous improvement of antioxidants and its components in jujube aimed at quality breeding program. A significant positive association between traits helps in improving several traits, simultaneously. Relationships between different biochemical traits of jujube fruit in current research suggests that selection for one of them could negatively or positively influence the amount of the other. The present study reports a strong positive correlation between the total phenolic content (mg GAE g$^{-1}$ fresh sample) and antioxidants (µmol g$^{-1}$ in FRAP). Similar relations between total phenols and antioxidants were also reported by Dudonne et al. [68]; Piluzza and Bullitta [69]; Aryal et al. [70]; Osman et al. [71] and Biswas et al. [72] in different plant species. Further, ascorbic acid content in the jujube fruit extracts positively correlated with their total phenols and antioxidants activity. The results are in agreement with those of Samee et al. [73] in 28 fruit extracts and Du et al. [74] in eight kiwifruit genotypes. Thus, it is plausible to suggest that phenols and/or ascorbic acid are primarily responsible for antioxidants activity in jujube fruit extract and that the presence of a substantial amount of total phenols in *Z. oenoplia* accession IC 625864 and of ascorbic acid in *Z. mauritiana* accessions IC 625849 may contribute significantly to antioxidants properties.

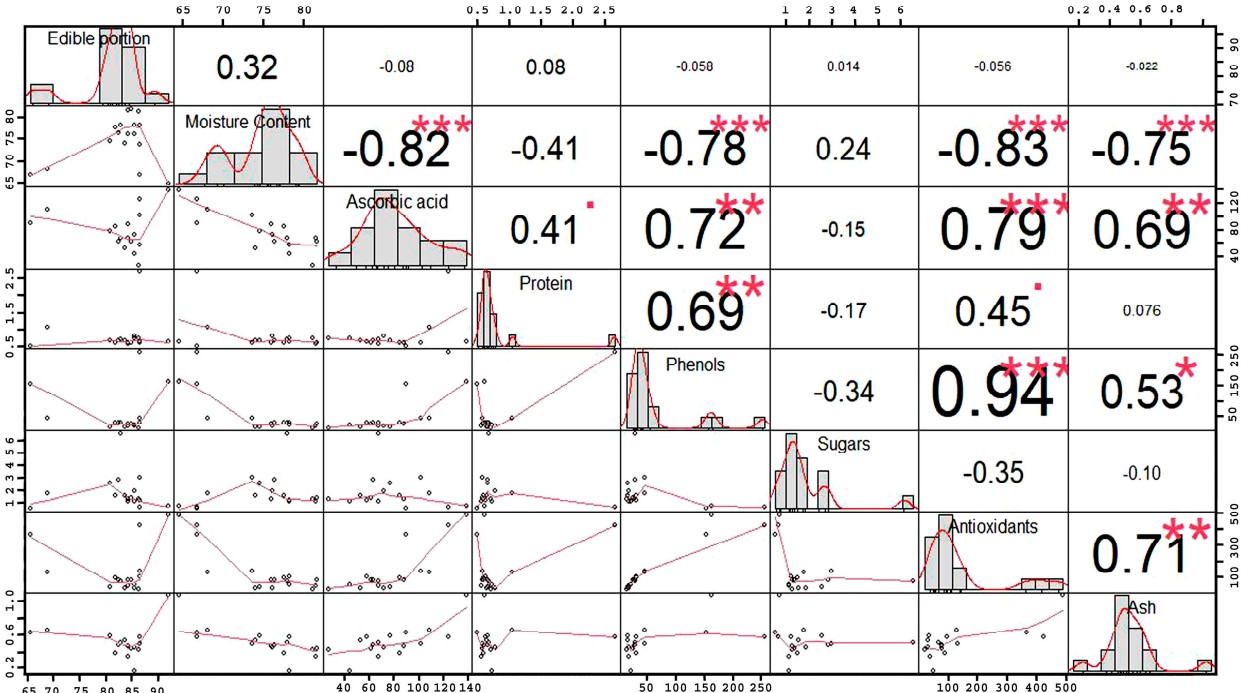

**Figure 4.** Correlation coefficients among biochemical parameters of jujube fruit quality (Edible portion, Moisture content, Ascorbic acid, Protein, Phenols, Sugars, Antioxidants and Ash) of the 18 genotypes. *, **, *** Significant at *p* < 0.05; < 0.01; < 0.001, respectively.

## 4. Conclusions

The study pertaining to an ex-situ collection of eighteen diverse Indian jujube accessions from arid and semi-arid areas, reveal that the IC 625864 (*Z. oenoplia*) found an extraordinary accession having fruits with very high phenols and total antioxidants (FRAP) contents while comparing with others. Moreover, IC 625849 (*Z. rotundifolia*) and IC 625848 (*Z. mauritiana*) were other genotypes contained high contents of phenols, ascorbic acid and total antioxidants. While aiming for simultaneous improvement program for total antioxidants with phenols, IC 625848, IC 625849 and IC 625864 can be considered useful genotypes, as also presented by a strong positive relationship between these traits. The genotypic and phenotypic coefficients of variations, heritability and genetic advance analysis in jujube fruits reveal that phenols, ascorbic acid and antioxidants contents are stronger and more stable quality traits in nature, thus could serve as a reliable biochemical marker to identify antioxidants rich genotypes. Hence, these can serve a basis of varietal selection in breeding programs aimed at producing fruits rich in functional properties. Information of characterization of jujube genotypes can be further employed for gene pool conservation and varietal improvement for future breeding programs. Traditional breeding methods have significant limitations when it comes to tree improvement, but newer technologies like tissue culture, plant genetic engineering to create transgenic cultivars, and molecular breeding techniques have some potential to combine different desirable and quality traits in the jujube.

**Author Contributions:** Conceptualization, V.S.M.; Designing of the experiments, A.K.; Contribution of experimental materials and Execution of field/lab experiments and data collection, R.B., K.S. and N.S.; Analysis of data and interpretation, V.S.M., V.G., K.R., P.K. and A.S.; Preparation of the manuscript, V.S.M., N.S., K.S., H.L., K.R., P.K., J.S.G. and A.S.; Review and Editing, P.K., A.S., J.S.G., V.G., A.K. H.L. and K.R. All authors have read and agreed to the published version of the manuscript.

**Funding:** This research received no external funding.

**Data Availability Statement:** Data related to study are provided in the paper.

**Acknowledgments:** Authors duly acknowledge support and facilities provided by Director, IC AR—NBPGR for carrying out study and encouraging during course of study.

**Conflicts of Interest:** The authors declare no conflict of interest.

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
