# Peer review of "Assessment of Genetic Variability for Fruit Nutritional Composition in the Ex-Situ Collection of Jujube (Ziziphus spp.) Genotypes of Arid Regions of India"

_horticulturae, doi:10.3390/horticulturae9020210_

Round 1

Reviewer 1 Report

This study holds scientific values for Jujube breeding by phenotypic assessment of several fruit traits in a Jujube germplasm collection. However, many issues need to be addressed for publication.

Firstly, an extensive correction on grammar and writing errors is needed.

A brief introduction of the status of jujube breeding should be added. Basically, why those morphological and biochemical characteristics were chosen for study? Is any of those traits related to Jujube’s adaptation to arid conditions. Please make connections to what you studied in this work, not just listing all the facts in the Introduction.

Line 123 How many trees are those three random trees chosen from?

Line124 What is the distinctness, uniformity, and stability (DUS) criteria? Please briefly explain or cite related literature.

Line 137 Which traits are quantitative? Which are qualitative?

Line 165 The author claimed the experiment was carried out in a randomized block design without providing any details. I highly doubt that is feasible in their study. It is an unsupported claim which indicates the authors have no idea what a randomized block design is.

Table 3 Is IC625863 a good representative of Z. mauritiana? The inter-species comparison is not well grounded as there are only one or two accessions from the other two species.

Table 4 Please provide the full name of TSS in the table and in the text.

Author Response

Dear Reviewer,

Below you may find the point-to-point response to each comment/ suggestion made in the review. We have revised the manuscript considering the comments of all the reviewers. The revision is made in track changes.

This study holds scientific values for Jujube breeding by phenotypic assessment of several fruit traits in a Jujube germplasm collection. However, many issues need to be addressed for publication.

Response: Authors are grateful to the reviewer for appreciating our work, and also for sparing time and providing valuable suggestions that helped to improve our manuscript quality.

Comment 1: Firstly, an extensive correction on grammar and writing errors is needed.

Response: Suggestion is accepted. The extensive revision has been done in view of all the suggestions, including the language part in the revised version of the manuscript.

Comment 2: A brief introduction of the status of jujube breeding should be added. Basically, why those morphological and biochemical characteristics were chosen for study? Is any of those traits related to Jujube’s adaptation to arid conditions. Please make connections to what you studied in this work, not just listing all the facts in the Introduction.

Response: The introduction is suitably modified, considering the reviewers' suggestions. We chose important fruit physical and biochemical traits to decipher the variability (at genotypic and phenotypic levels) among the jujube accessions of arid and semi-arid regions used in the study. The superior genotypes were identified based on targeted fruit nutritional and functional properties which could be considered for the breeding program aimed at fruit quality enhancement, and also for direct use. Ber possesses some peculiar characteristics that help adapt trees under hot arid climates; this has already been described in a review paper by our group (Meena et al., 2022; reference 2 in MS); here, our main emphasis was on fruit physical-biochemical traits, and which are adequately described.

Comment 3 (Line 123): How many trees are those three random trees chosen from?

Response: The sentence is corrected and better explained. Three trees, one in each replication, for each accession were considered in the study from which 20 randomly harvested fruit samples were taken for physical-biochemical analysis.

Comment 4 (Line 124): What is the distinctness, uniformity, and stability (DUS) criteria? Please briefly explain or cite related literature.

Response: Thanks for the point about DUS. We have provided an explanation here, but we decided to remove those sections from the paper, considering the reviewer's comments and suggestions (comment 6). “The distinctness, uniformity, and stability (DUS) is provided for the protection of new plant varieties and recognition of plant breeders’ rights under the UPOV convention (1991) if it is distinct (D) from any other existing variety, sufficiently uniform (U), and stable (S) (Dixitet al. 2010).

Comment 4: Line 137 Which traits are quantitative? Which are qualitative?

Comment: We have described the traits, which are quantitative (measurable) and which are qualitative (non-measurable) in the revised version of the manuscript at the appropriate place.

Comment 5 (Line 165): The author claimed the experiment was carried out in a randomized block design without providing any details. I highly doubt that is feasible in their study. It is an unsupported claim which indicates the authors have no idea what a randomized block design is.

Comment: A total of three trees per accession (per replication) were considered in the study. The accessions were planted in a randomized fashion across the three replications. Hence, there are eighteen treatments and three replications in this experiment. Further, fruit analysis was done using 20 randomly sampled fruits from each tree/replication for each accession.

Comment 6: Table 3 Is IC625863 a good representative of Z. mauritiana? The inter-species comparison is not well grounded as there are only one or two accessions from the other two species.

Response: We agree with the comment of the reviewer on the valid point related to inter-species variation and comparison. Hence, we decided to remove table 3 and related text as this part is not much related to our aim which is related to the assessment of variabilities in fruit's physical-biochemical characteristics.

Comment 7: Table 4 Please provide the full name of TSS in the table and in the text.

Response: As suggested, TSS is elaborated to total soluble solids in Table 4, now table 3.

Reviewer 2 Report

The authors have described the Genetic Variability of different Jujube. However, there are some issues that need to clear before making it publishable such as-

1. Table 2: How pulp taste and aroma become morphological characteristics? I recommend revising the table title.

2. In Fig 1: Only two genotypes (IC625849 and 64) showed shorter fruit length than width. Please describe the reasons that might be involved here.

3. In Table 4: The authors summarized TSS, Ascorbic acid, Total Phenol, and Antioxidant content for 2020-21, and 2021-22 which are not significantly different. And these were the almost same in the previous years, if little difference, there are so many factors that might be involved. Therefore, why it is important to relate these parameters to the genetic variability of different Jujube genotypes? 

I would suggest revising the respective paragraphs and rewrite them considering the issues mentioned above before making it publishable. 

Author Response

Dear Reviewer,

Below you may find the point-to-point response to each comment/ suggestion made in the review. We have revised the manuscript considering the comments of all the reviewers. The revision is made in track changes.

Comment 1: Table 2: How pulp taste and aroma become morphological characteristics. I recommend revising the table title

Response: We agreed to a valid comment of the reviewer, but now this table is removed, considering the comment of reviewer #1 (comment 8).

Comment 2: In Fig. 1 only two genotypes (IC625849 and 64) showed shorter fruit length than width. Please describe the reasons that might be involved here.

Response: Thanks reviewer for the comment. We have inserted a description of why the fruit length can be shorter than the width and thus the peculiar shape of the fruit. [The distinctive phenotypic traits of the germplasm are controlled primarily by four major QTLs (genetic maternal) like globe, sun, ovate, and fs8.1. One of them, the globe, had a greater unique effect on fruit height than fruit width, and this is likely the primary factor driving changes in fruit size (proximal end) (Zhang et al., 2007; Sierra-Orozco et al., 2021). Thus, Ber fruit length may be shorter than width depending on fruit shape, which can range from obovate/round/ oval to oblong].

Comment 3: In Table 4. The authors summarized TSS, Ascorbic acid, Total Phenol, and Antioxidant content for 2020-21, and 2021-22 which are not significantly different. And these were the almost same in the previous years, if little difference, there are so many factors that might be involved. Therefore, why it is important to relate these parameters to the genetic variability of different Jujube genotypes? 

I would suggest revising the respective paragraphs and rewriting them considering the issues mentioned above before making it publishable. 

Comment: We partially agree with the reviewer's comments. TSS, Ascorbic acid, Total Phenol, and Antioxidant content for 2020-21, and 2021-22 are not significantly different, but our emphasis is on germplasm accessions, and these components are significantly different among genotypes as shown in ANOVA, which showed that germplasm among themselves was diverse for these traits. No significant seasonal (year) differences in these characters indicate that they are not significantly affected by the environment as shown by high heritability (%) together with high GCV and PCV. The required elaboration is added in the paragraph.

We thank the reviewer for the valuable suggestions that helped improve manuscript's quality.

Reviewer 3 Report

The abstract should be better summarized.

The authors should mark the importance to study the nutrition composition of food matrix representing the biodiversity and traditions of a place and related references added such as:

Durazzo, A. The Close Linkage between Nutrition and Environment through Biodiversity and Sustainability: Local Foods, Traditional Recipes, and Sustainable Diets. Sustainability 201911, 2876. https://doi.org/10.3390/su11102876

A graphical scheme of study approach should be inserted.

Results in Table 4 should be better described in the text.

Figure 4 should be better described in the text.

Limits, advantages and practical applications should be inserted in Conclusion.

Author Response

Dear Reviewer,

Below you may find the point-to-point response to each comment/ suggestion made in the review. We have revised the manuscript considering the comments of all the reviewers. The revision is made in track changes.

Comment 1: The abstract should be better summarized.

Response: As suggested by the reviewer, we have considerably improved the abstract.

Comment 2: The authors should mark the importance to study the nutrition composition of food matrix representing the biodiversity and traditions of a place and related references added such as:

Durazzo, A. The Close Linkage between Nutrition and Environment through Biodiversity and Sustainability: Local Foods, Traditional Recipes, and Sustainable Diets. Sustainability 2019, 11, 2876. https://doi.org/10.3390/su11102876

Response: The importance to study the nutrition composition is suitably inserted with reference in the introduction.

Comment 3: A graphical scheme of study approach should be inserted.

Response: Thank you for the comments. The study approach has been adequately described in the materials and methods section in the text which is self-explanatory.

Comment 4: Results in Table 4 should be better described in the text.

Response: Results are described in the text, and further elaborated in the results section.

Comment 5: Figure 4 should be better described in the text.

Response: Genetic correlations are described in the text in elaborated form.

Comment 6: Limits, advantages and practical applications should be inserted in Conclusion.

Response: The conclusion is considerably improved, as per the suggestions.

Sincere regards

All the reviewers

Round 2

Reviewer 1 Report

I appreciate the authors' response. However, some revisions are still not clear to solve the original issue.

The added detail in Line 161-163 suggests the conducted experiment was not based on a randomized block design. A randome selection during sampling is not called as randomized block design (line 160).

Quantitative and qualitative traits are not differentiated based on whether they are measurable or not.

I sincerely hope the authors should do more homework on these common terms in genetics for this manuscript.

Author Response

Dear Reviewer,

Please find below the response to each comment/ suggestion that has been incorporated into the current form of the manuscript.

Comments: I appreciate the authors' response. However, some revisions are still not clear to solve the original issue.

The added detail in Line 161-163 suggests the conducted experiment was not based on a randomized block design. A random selection during sampling is not called as randomized block design (line 160).

Response: We agree with the valid comment highlighted by the reviewer. Sorry for not properly addressing this issue. We had provided the justification in the response sheet but missed incorporating it in the manuscript. This was in RBD, because one tree for each of the eighteen accessions (treatments) was planted randomly in each of the three blocks (replications). Thus, each block (replication) has eighteen treatments. For physical-biochemical analysis, twenty fruits were randomly sampled from each tree (replication). Regarding random fruit sampling, we actually meant randomly sampled fruits (from different branches/ sides) to minimize sampling error not randomization per se.

We have suitably modified the statements in the revised version.

Comment: Quantitative and qualitative traits are not differentiated based on whether they are measurable or not.

Response: Yes, we agree with that, these are considered to be quantitative or qualitative based on variation (continuous-quantitative/ discontinuous-qualitative), gene(s) (mono/oligo-qual. / several-quant.), stability (low-quant. / high-qual.) and so on. Thanks for your remarks. In the revision, the traits are suitably designated as fruit physical (e.g., fruit length, width, etc.) and fruit biochemical (e.g., TSS, phenol, ash, etc.) traits which are commonly used in studies related to horticultural plants concerning the fruit quality.

I sincerely hope the authors should do more homework on these common terms in genetics for this manuscript.

Thanks for the suggestions and for sparing valuable time to review and provide valuable suggestions that helped improve manuscript's quality.

Sincere regards

from all the authors

Reviewer 2 Report

The authors revised the manuscript considering the concerns. I would like to recommend modifying the overall sentences to make clearly understandable before publishing.

Author Response

Dear Reviewer,

Thanks for sparing valuable time to provide valuable suggestions which helped improved manuscript quality significantly.

We have provided the response to the suggestion and same has been implemented in the current version of manuscript. 

Comment: The authors revised the manuscript considering the concerns. I would like to recommend modifying the overall sentences to make clearly understandable before publishing.

Response: Dear Reviewer, sorry for the inconvenience that has probably occurred and caused difficulty to follow the sentences due to extensive revision made in track change mode. In the current form, after also considering the comments of another reviewer, we have suitably modified the sentences which are now clear and understandable. We hope this revised version is ok. Any minor error, if noticed, will be taken care of during proofreading, if MS is accepted.

Thanks and regards

from all the authors

Round 3

Reviewer 1 Report

I appreciate the authors' response. I don't have further comments.

Author Response

Dear Reviewer,

Thanks a lot for your remarks and valuable inputs to improve MS quality